# Effect of Static Magnetic Field of Electric Vehicles on Driving Performance and on Neuro-Psychological Cognitive Functions

**DOI:** 10.3390/ijerph16183382

**Published:** 2019-09-12

**Authors:** Yaqing He, Weinong Sun, Peter Sai-Wing Leung, Yuk-Tak Chow

**Affiliations:** 1Department of Electrical Engineering, City University of Hong Kong, 83 Tat Chee Avenue, Hong Kong, China; wnsun2-c@my.cityu.edu.hk (W.S.); or eeytc@cityu.edu.hk (Y.-T.C.); 2EMC Consortium Limited, Hong Kong, China

**Keywords:** static magnetic field (SMF), electric vehicles (EVs), driving performance, neuro-psychological cognitive functions

## Abstract

Human neuropsychological reactions and brain activities when driving electric vehicles (EVs) are considered as an issue for traffic and public safety purposes; this paper examined the effect of the static magnetic field (SMF) derived from EVs. A lane change task was adopted to evaluate the driving performance; and the driving reaction time test and the reaction time test were adopted to evaluate the variation of the neuro-psychological cognitive functions. Both the sham and the real exposure conditions were performed with a 350 μT localized SMF in this study; 17 student subjects were enrolled in this single-blind experiment. Electroencephalographs (EEGs) of the subjects were adopted and recorded during the experiment as an indicator of the brain activity for the variations of the driving performance and of the cognitive functions. Results of this study have indicated that the impact of the given SMF on both the human driving performance and the cognitive functions are not considerable; and that there is a correlation between beta sub-band of the EEGs and the human reaction time in the analysis

## 1. Introduction

With the exponential increase in popularity of electric vehicles (EVs) in transportation, the electromagnetic fields (EMFs) of EVs have aroused increasingly public concerns. The EMFs inside the vehicles are either from vehicle drivers, and passengers exposed to long-term EMFs inside EVs is a considerable safety issue. The EMFs are induced by the electric powertrains [1,2,3,4,5,6], or by a portion of the electric system such as in the power cables [7], inverters [8], or batteries [6,9] of EVs; or from wireless communication systems [10,11]. EMF distributions from EVs, either time-varying EMFs [2,3,6,9], static magnetic fields (SMF) (including DC scenarios) [8,9], or synthesized scenarios of above two, have been evaluated by both measurements and computational simulations. The EV-derived EMFs are in general in compliance with the basic restriction values of international standards [1,2,6], however, some EMFs levels in the scenarios in which the other parameters are in compliance with the basic restrictions may exceed the reference limits, threatening the safety of human health in EVs [3,4].

The reference limits are well established by the International Commission on Non-Ionizing Radiation Protection (ICNIRP), for preventing acute effects of EMF exposure either on the heating effects to body tissues or on the electro-stimulations to transient nervous system responses [12,13,14]. However, the effects prescribed in terms of in-body parameters are not easy to determine, especially in the central nervous system (CNS). In the field of bioelectromagnetics, the effects have yet to be fully characterized on cells, tissues, whole-body animal subjects or human volunteers [15].

EMFs, specifically the static magnetic fields (SMFs), will affect human neuro-psychological functions in eye–hand coordination [16], short-term memory and attention [17], the visual function [16,18,19], and the reaction time [17,19,20]. It has also been pointed out that negative effects on human neuro-psychological functions may lead to distraction driving [21] where the attention of the drivers is distracted away from the road and has been concluded as a factor leading to collisions and near misses [22], which have not been fully addressed.

This paper investigates the effect of SMF of EVs on driving performance as well as on cognitive functions. The lane change task (LCT) test was adopted for the driving performance evaluation; the driving reaction time (DRT) test and the reaction time (RT) test were for driving response and for intuition response, respectively, of the cognitive function evaluation. Electroencephalographs (EEGs) were adopted as a physiological indicator of the brain functional activities in the study, EEG sub-bands were then classified into different cognitive functions reflecting the brain activities [23,24].

The hypothesis of this study is that SMF may affect the driving performance and the cognitive functions, with the considerations of the assessment of the driving performance, of the driving reaction time and of the intuition reaction time. The EEG oscillations are also measured during the experiment with further correlation to the driving performance, the driving reaction time and the reaction time, under exposure.

In this paper, the work is organized as follow: in the Experimental Set-Up section, the experimental concept, design, the evaluation methods for driving performance and cognitive functions, as well as the statistical analyzing methods are introduced in detail. The results achieved by the statistical analyses are presented in the Result session, and correspondingly discussed in the Discussion session. The conclusion are drawn in the last section.

## 2. Experimental Set-Up

The experiments were conducted in an EMC anechoic chamber for achieving a clean electromagnetic controlled environment, in the Applied Electromagnetic Laboratory of the City University of Hong Kong. The experimental set-up in the chamber is shown in Figure 1.

The experiment was established as a repeated-measurement experiment [15] evaluation of the driving performance DRT, and RT with the sham exposure condition conducted, and then repeated with the SMF exposure condition, for comparing their differences.

### 2.1. Subject Selection

A total of 17 students (average age = 22.01, standard deviation = 0.84) volunteered as subjects for this study. The students were healthy undergraduates from the City University of Hong Kong, and the experiment was conducted in accordance with the human ethics review approval (H000622) from the Human Subjects Ethics Sub-Committee of the City University of Hong Kong. Previous driving experience was not compulsory. Exclusive criteria for the subjects in the experiment was the presence of metallic implant objects, taking substances affecting mental health, e.g., psychotropic drugs, sedatives or alcohol in the last 24 h, any cardiovascular, neurological, or psychological disorder. Sufficient sleep was required before the experiment.

### 2.2. Static Magnetic Field (SMF) Exposure

An SMF with intensity of 350 μT was adopted for the experiment, which is above the highest value from the a literature survey of 300 μT [6]. A solenoidal coil was fabricated for generating the required SMF, having two layers with a diameter of 25 cm; each layer consisted of 22 turns of copper wire 0.27 mm in diameter, as in Figure 1. The SMF intensity can be directly calculated with Biot-Savart’s Law [25]. The temperature rise of the coil has been established by measurement to be at room temperature of a maximum of 40 °C with 1.5 A current input in the experiment.

### 2.3. Evaluating Methods

#### 2.3.1. Lane Change Task (LCT) Test 

Lane change task (LCT) was adopted in this study. The LCT method is developed for distraction-driving assessment [26] and has been verified as easy-to-use, reliable, and efficient [27,28,29]. The score of the driving performance in LCT is evaluated in terms of the mean deviation introduced in [30], as illustrated in Figure 2. In the experiment, the driving simulator set-up, includes a computer with driving simulation software (OpenDS 4.0), (German Research Center for Artificial Intelligence, Saarbrücken, Germany), a steering wheel, and a braking and accelerating pedal, in compliance with the ISO requirements [26]. The LCT test route is a straight three-lane road of 3 km with 18 lane-changing signs. All lane-changing signs are about 150 m apart, along the route and on both sides of the road, and will pop up 40 m before each lane change. Subjects are required to implement the lane-change manoeuvres while maintaining 60 km/h as a steady speed; the LCT test will last for about 3 min.

#### 2.3.2. Driving Reaction Time (DRT) Test

The DRT test was conducted to estimate the reaction time during driving. Subjects are required to drive along a 2 km straight road at a speed of 80km/h. There are 7 red blocks in total, popping up 50 m randomly on the lanes before the vehicle approaches each block, as illustrated in Figure 3. The subjects are asked to brake immediately when the red blocks are noticed. Time between each red block appears, and the subject pressing the brake paddle is timed by the computer with a timer program and is defined as the “driving reaction time”. The average of the shortest 5 time records will be used for analysis.

#### 2.3.3. Reaction Time (RT) Test 

Subjects are required to sit in front of the computer instead of the driving simulator, keeping the left mouse button pressed, as illustrated in Figure 4. They are required to release the button immediately when a white dot shows up on the monitor. The reaction time will be recorded automatically with the computer timer program for 7 times, and the average of the shortest 5 times will be used for analysis. This test will take about 2 min.

### 2.4. Experimental Procedure 

The experiment was conducted as a single-blind test experiment, meaning that the subject did not know when the SMF was applied. There are 17 subjects volunteered participating in this experiment. They were briefed with the experiment information and were required to complete a consent form, to wear an EEG cap, and provided with a 4-miunte free driving period for familiarizing with the driving simulator. The LCT test, the DRT test and the RT test were then conducted in turn, and the EEGs were also collated simultaneously. All of the LCT tests, the DRT tests and the RT tests were conducted under the sham exposure condition of no SMF exposure, and then repeated under real exposure of 350 μT. A 5-min break was given between two conditions for each subject. 

### 2.5. Data Analysis 

The scores of the LCT were collected for indicating the driving performance; the reaction times in the DRT test and the RT test were collected for cognitive functions. They were all statistically analyzed with the software of R Programming Language 3.5.3, with a confidence interval of 95%. 

For the hypothesis analysis of this study, the normality of the results were firstly verified by the Shapiro–Wilk test; paired t-test was used to determine the measurand conforming to the normal distribution, otherwise Wilcoxon signed rank-sum test was adopted by comparing the outcomes of the sham and the real SMF exposure. A multivariate analysis of variance (MANOVA) was then carried out, with a factor of three EEG sub-bands (theta, alpha and beta) and another factor of two exposure conditions (sham and real), for the impact of the SMF to the brain functional activities with the EEG power spectrum density (PSD) data. A two-way analysis of variance (ANOVA) for correlating the EEG PSD to the driving performance and cognitive performance was also performed. The two-way ANOVA is with a factor of two SMF exposure levels (sham and real) × a factor of three EEGs sub–band levels (theta, alpha and beta). Regression analysis was carried out to identify any significant correlation of either driving performances or cognition functions with the EEG oscillations.

## 3. Results

### 3.1. Analysis of Driving Performance and Cognitive Function Results

Wilcoxon signed rank-sum test was applied for the LCT and RT results (LCT: *W* = 55, *p* = 0.33; RT: *W* = 39.5, *p* = 0.15), while paired t-test was applied for the DRT result (t = 0.30, *p* = 0.77). No significant difference was observed between the sham and real SMF exposure conditions in all tests of LCT, DRT and RT.

### 3.2. Impact of SMF on Electroencephalographs (EEGs)

PSD data of the Delta sub-band obtained in the FFT of the EEG analysis were all zeros, and hence the Delta sub-band was not further considered in subsequent data processing. This awake stage of the driver in the experiment is in corresponding to the inactive characteristics of the Delta sub-band [31].

The results of the one-way MANOVA are presented in Table 1. The outcome indicates that there was no effect of SMF on other simultaneous EEG sub–band oscillations.

### 3.3. Effect of Interaction of SMF on EEGs with Driving Performance and Cognitive Functions

The outcome of data analysis of the interaction of SMF and the EEGs with driving performance and the cognitive function performances is illustrated in Table 2.

As Table 2 illustrates, there is only one significance being observed in the RT test on the beta sub-band (F = 5.961, *p* = 0.0217, Partial η^2^ = 0.19, Power = 0.18). The significance level (*p* = 0.0217) indicates that the impact of beta sub-band fluctuation on subjects’ reaction time is considerable. No other significant differences were obtained in the ANOVA analysis results for the tests of LCT, DRT and RT.

The results of the regression analysis of the reaction times and the simultaneous beta PSD data under different exposure conditions are illustrated in Table 3. It was observed that there is a significance of the relationship between the reaction time and the beta sub-band.

A marginal significant correlation (*p* = 0.0684) of beta PSD under the sham exposure condition, and significant correlation (*p* = 0.0182) of beta PSD under the real exposure condition were noted in Table 3, indicating that there is a positive correlation of the beta sub-band with the reaction time in all conditions.

## 4. Discussion

This pilot study examines the effect of SMF of EVs on driving performance and on neuro-psychological cognitive functions. The driving performance of 17 subjects was assessed by the LCT test, and the cognitive functions of driving reaction time and reaction time of them were assessed by the DRT test and the RT test.

### 4.1. Discussion of the Relationship between the Beta Sub-Band and RT

It was observed in Table 2 that there is a significant effect of the beta sub-band to the reaction time, and then Table 3 demonstrates that there are two positive correlations, under both sham and real exposures, between the beta sub-band and the reaction time in the regressive analysis. The correlation under the sham exposure is of a marginal significance with a beta weight of 6.952 × 10^−6^, an adjusted R^2^ of 0.1515, and a *p* value of 0.0684, implying that the correlation between the beta oscillation and the reaction time is considered relatively weak. The other correlation under the real SMF exposure is significant with a beta weight of 8.852 × 10^−6^, an adjusted R^2^ of 0.2733, and a *p* value of 0.0182, implying a strong correlation between the beta oscillation and the reaction time.

The beta-band brain activity is related to the maintenance of the human cognitive states [32,33]. Negative correlations have been reported in some studies [33]. Our experiment concluded a positive correlation between the beta oscillations and the reaction times. Either a positive or a negative correlation confirms that there is a direct correlation between the beta wave responses and the reaction time.

### 4.2. Discussion on the Results of the DRT and the LCT

It is considered that the reaction time is the intuition reaction, while the DRT is the reaction as a response under driving situation. In the DRT test, no significant effects were observed in Table 2 under any of the conditions of SMF, the isolated sub-band waves, or any interactions of the SMF and the sub-bands. This observation indicates that the reaction time would not be affected by the given SMF. The same findings, i.e., no significant effects, could also be obtained from the results of LCT in Table 2 for the driving performance. The insignificant outcomes might be due to the fact that the SMF exposure time in our experiment is relatively short as compared to the long exposure of drivers in EVs. A short exposure time may not lead to the establishment of a “cumulative effect” [34]. 

### 4.3. Discussion of the Electromagnetic Field (EMF) 

No significant changes of the records on either driving performance (LCT: *W* = 55, *p* = 0.33) or on cognitive assessment (DRT: *t* = 0.30, *p* = 0.77; RT: *W* = 39.5, *p* = 0.15) was observed, indicating that there is no immediate considerable impact of SMFs derived from EVs upon the driving performance and cognitive functions. These outcomes are in line with other research investigations of different exposure intensities and of different applications [16,17,19,20]. There is also no direct significant impact of SMF observed on the EEG data in our study, as in Table 1, demonstrating that there is no considerable impact of SMFs to human brain activities during driving. 

The intensity of SMF of 350 μT associated with EVs’ has been adopted in our study, which is relatively lower than the SMF exposures of up to 8 Tesla in magnetic resonance imaging/functional magnetic resonance imaging(MRI/fMRI) applications [16,17,19]. It was observed that there are significant impacts of near visual contrast sensitivity [17], a negative tendency of SMF to eye-hand coordination [18], yet no significant impact of other cognitive functions due to SMF of MRI applications [16,17]. No conclusive results could be drawn from recent studies, and no relevant studies have addressed the low field strength in EV applications. It is the intension of this paper to address the level of magnetic field in the EV applications in terms of driving performance and cognitive function for driving safety. Therefore, it is reasonable to expect that no significant influence for the low intensity SMF exposure derived from EVs to human driving performance and cognitive functions. 

F. de Vocht et al. have pointed out that the effects on cognitive functions may depend on the time-varying fields [19], and the time-varying low-frequency EMF has been regarded to have adverse risk to the central nervous system and the peripheral nervous system [34]. There are, however, insufficient outcomes to draw a decisive conclusion on the adverse risks of the time-varying EMFs. The adverse risk of time-varying EMF to drivers is not covered in this paper.

## 5. Conclusions

This pilot study provides a study correlating the effect of SMF derived by EVs to human driving performance and cognitive functions in terms of the LCT test, DRT test and RT test. The performances of driving performances and cognitive functions correlated with subjects’ simultaneous EEGs fluctuations. Results of this study have indicated that the impact of the given SMF on both the human driving performance and the cognitive functions are not considerable; results have also suggested that there is a correlation between beta sub-band of the EEGs and the human reaction time in the analysis. 

## Figures and Tables

**Figure 1 ijerph-16-03382-f001:**
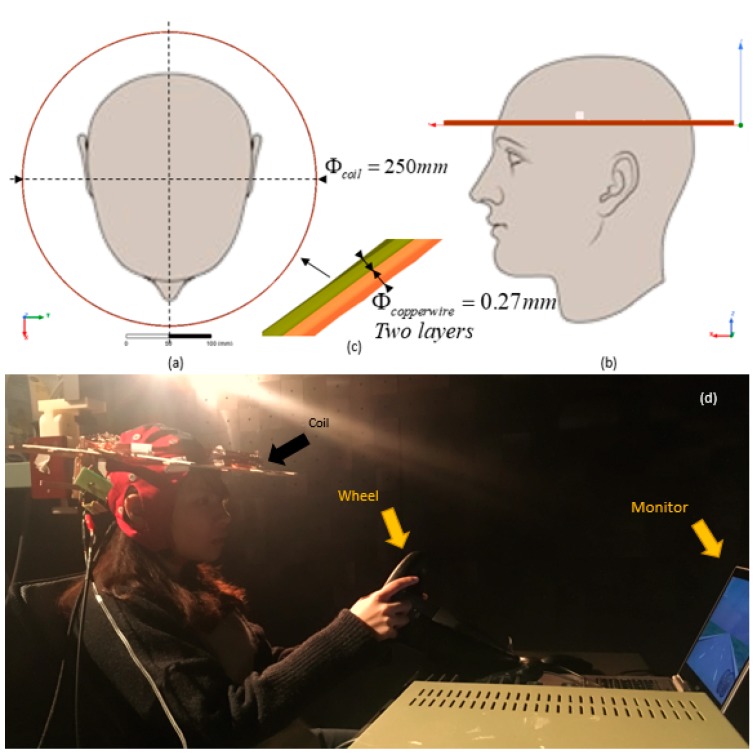
The set-up of the experiment in an anechoic chamber. (**a**) the top view and (**b**) the left view elaborate the relative position between subjects’ head and the coils, respectively; (**c**) elaborates the two-layer structure, as well as the parameter of the coil in the experiment; (**d**) demonstrates the overview of the setting-up of the experiment in the anechoic chamber.

**Figure 2 ijerph-16-03382-f002:**
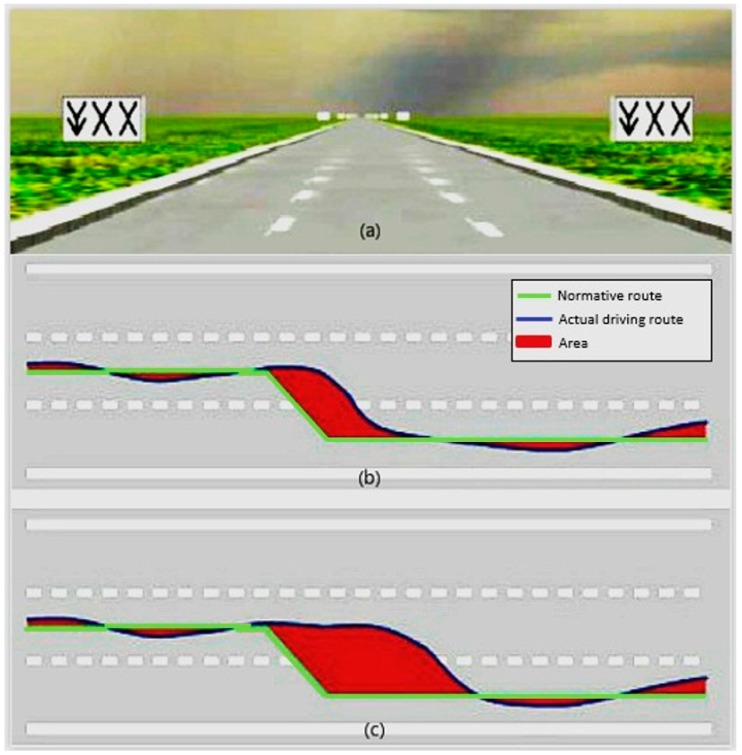
(**a**) illustrates the screen view of lane change task (LCT). (**b**,**c**) illustrates the difference in area (red) between the normative path (green line) and actual path(blue line) for a quick perception adapted from [26]. The area in (**b**) is smaller than that in (**c**), which demonstrates the performance of lateral positioning in (**b**) is greater than that in (**c**).

**Figure 3 ijerph-16-03382-f003:**
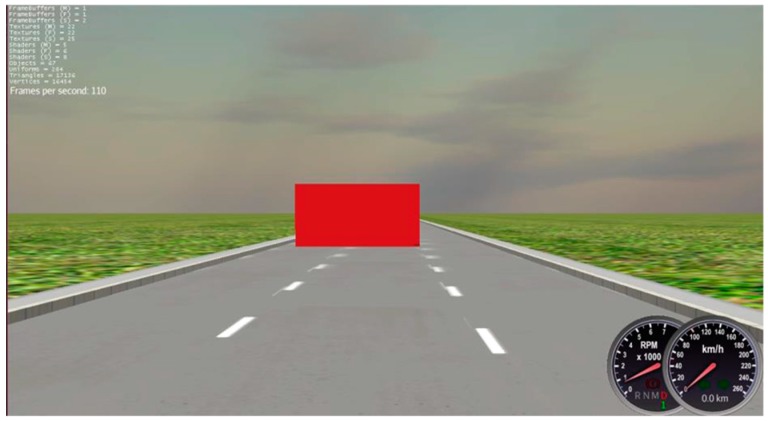
Demonstration of the operating system for driving reaction time (DRT) test adopted in this experiment.

**Figure 4 ijerph-16-03382-f004:**
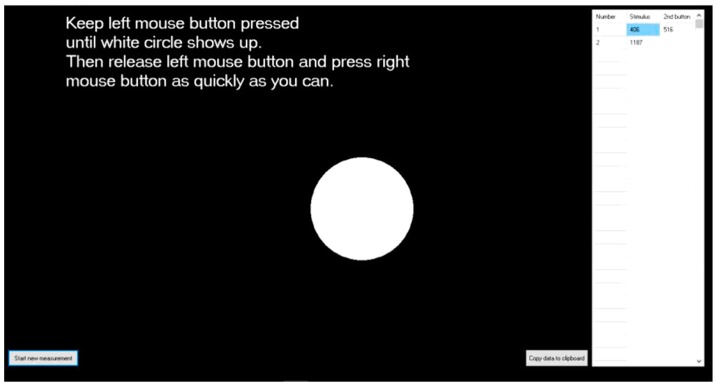
Screen capture of reaction time (RT) test.

**Table 1 ijerph-16-03382-t001:** Results from the one-way multivariate analysis of variance (MANOVA) with repeated-measurement design of power spectrum density (PSD) data of electroencephalograph (EEG) sub-bands.

Test	theta	alpha	beta
LCT	F = 0.9827	F = 0.9808	F = 0.7262
	*p* = 0.3290	*p* = 0.3294	*p* = 0.4005
	Partial η^2^ = 0.03	Partial η^2^ = 0.03	Partial η^2^ =0.02
	Power = 0.17	Power = 0.17	Power = 0.14
DRT	F = 0.0625	F = 0.0063	F = 0.0007
	*p* = 0.8042	*p* = 0.9373	*p* = 0.9786
	Partial η^2^< 0.01	Partial η^2^< 0.01	Partial η^2^< 0.01
	Power = 0.0572	Power = 0.0507	Power = 0.0501
RT	F = 0.1301	F = 0.1139	F = 0.0740
	*p* = 0.7207	*p* = 0.7380	*p* = 0.7873
	Partial η^2^ < 0.01	Partial η^2^ < 0.01	Partial η^2^ < 0.01
	Power = 0.07	Power = 0.06	Power = 0.06

(“Partial η^2^” refers to the “effect size”, and “Power” refers to the statistical power of the analysis. *p* < 0.05).

**Table 2 ijerph-16-03382-t002:** Results of two-way analysis of variance (ANOVA) with repeated-measurement design of score from lane change task (LCT), driving reaction time (DRT) and reaction time (RT) tests for static magnetic field (SMF) exposure, the average PSD values of theta, alpha and beta sub-bands, and the Inter. A (interaction of SMF exposure x theta), Inter. B (interaction of SMF exposure x alpha) and Inter. C (interaction of SMF exposure x beta) in three tests.

	LCT				DRT				RT			
	F	*p*	P. η^2^	Pwr.	F	*p*	P. η^2^	Pwr.	F	*p*	P. η^2^	Pwr.
**SMF**	0.0061	0.9381	<0.01	0.05	0.0072	0.9332	<0.01	0.05	1.1799	0.28735	0.04	0.07
**theta**	0.0264	0.8721	<0.01	0.05	0.4533	0.5067	0.02	0.05	1.6363	0.2121	0.06	0.06
**alpha**	0.0124	0.9122	<0.01	0.05	0.3192	0.5769	0.01	0.05	1.722	0.2009	0.06	0.06
**beta**	0.3172	0.5781	0.01	0.05	0.2879	0.5961	0.01	0.05	5.961	0.0217 *	0.19	0.18
**Inter. A**	0.0120	0.9137	<0.01	0.05	0.0052	0.9429	<0.01	0.05	0.0631	0.8036	<0.01	0.07
**Inter. B**	0.0158	0.9008	<0.01	0.05	0.0059	0.9391	<0.01	0.05	0.0381	0.8467	<0.01	0.06
**Inter. C**	0.0112	0.9165	<0.01	0.05	0.0005	0.9825	<0.01	0.05	0.1148	0.7374	<0.01	0.08

(“P. η^2^” refers to “Partial η^2^”, and “Pwr.” refers to the power of the analysis. “*” highlights *p* < 0.05).

**Table 3 ijerph-16-03382-t003:** Regression findings for the results of RT assessment with simultaneous beta PSD data under different exposure conditions, where “Expo. Con.” refers to “Exposure Conditions”, and “Std Err.” refers to “Standard Errors”.

Expo. Con.	Beta Weight	Std Err.	Multiple R^2^	Adjusted R^2^	*p* Value
Sham	6.952 × 10^−6^	3.540 × 10^−6^	0.2045	0.1515	0.0684 #
Real	8.852 × 10^−6^	3.342 × 10^−6^	0.3187	0.2733	0.0182 *

(“*”: *p*< 0.05; “#”: *p*< 0.1)

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
