# Peer review of "Effect of Static Magnetic Field of Electric Vehicles on Driving Performance and on Neuro-Psychological Cognitive Functions"

_ijerph, 2019, doi:10.3390/ijerph16183382_

Round 1
Reviewer 1 Report
This work evaluates the influence of the magnetic field exposure on the neurological reactions of drivers. The work is correct but you should take into account the followiing recommnedations:
1.- The work is correct, but I think that it should be reviewed by a native english.
2.- Thare are some typo errors:
3.- Be careful with the acronyms, in the abstract, it is not necessary to indicate the acronyms for the expressions that are not repeated in the own abstract.
4.- You should resume the different sections fo the paper at the end of the introduction section
5.- The inteerest of the topic of work should be highlighted in the Introduction. You should remark the relevance of the study of human exposure in vehicles [*]:
[*] Celaya-Echarri, M; Azpilicueta, L; Lopez-Iturri, P; Aguirre, E; De Miguel-Bilbao, S; Ramos, V; Falcone, F. "Spatial Characterization of Personal RF-EMF Exposure in Public Transportation Buses," in IEEE Access, vol. 7, pp. 33038-33054, 2019.

Reviewer 2 Report
Effect of Static Magnetic Field of Electric Vehicles on Driving Performance and on Neuro-Psychological Cognitive Functions
Yaqing HE, Weinong SUN, Peter Sai-Wing LEUNG, and Yuk-Tak CHOW
The authors present a new study on the influence of small static magnetic fields on driving performance and cognitive functions.
To my point of view the Details are well presented, the discussion and the conclusions are sound.
Reading the article in the present form there were some questions I got. Maybe the authors can add some more information. Also within the discussion some aspects on the strength of magnetic fields might be added.
Therefore I recommend the article for publication after minor revision.
Line 39ff: Here the publications 15-19 are presented. From the title of these publication it seems that the influence of high magnetic inductions above 1Tesla was applied.
Within the current work an induction of 350 micro-Tesla was applied – about a factor 10 higher than the static earth magnetic field, but several orders of magnitude smaller than in the cited publications.
I recommend adding a discussion on the influence of the field-strength.
Time varying EMFs are not covered within the introduction. But these effects are presented within the discussion chapter lines 227ff. To my point of view the lines 227ff should be shifted to the introduction part. The effect of heating by time varying magnetic fields depends on amplitude and frequency. Discussions of time varying EMFs on health features are going on.
Line 69, line 125ff: “Previous driving experience was not compulsory” – To my point of view driving skills depend on some experience. Was there an influence within the results of experienced drivers and less experienced ones?
According to the procedure (line 125ff) each subject performed the test under sham exposure first. Then it was repeated under field exposure. Assuming that the experience from the first run improves the subjects driving ability this could lead to a compensation of negative effects of the magnetic field.
Why is this assumption from me not correct?
Figure2: the green line in the legend cannot be seen very well. I recommend to optimize the figure.
Lines 117 to 124 are repeated in lines 125 to 133.
Line 157: “ … This awake stage of the driver in the experiment in corresponding to the inactive characteristics of the ...” maybe an “in” should be an “is”: “experiment is corresponding”
Line 180 “…Figure 3, indicating that ...” – I guess it should be “table 3”
Line 222 ff: Here the authors present that in previous work inductions in the range of Tesla did not lead to “significant impact”.
I strongly recommend to shift the presentation of different field strength to the introduction and give a motivation why the authors investigated the small magnetic fields.
Author Response
The authors present a new study on the influence of small static magnetic fields on driving performance and cognitive functions.
To my point of view the details are well presented, the discussion and the conclusions are sound.
Reading the article in the present form there were some questions I got. Maybe the authors can add some more information. Also within the discussion some aspects on the strength of magnetic fields might be added.
Therefore I recommend the article for publication after minor revision.
Point 1(1): Line 39ff: Here the publications 15-19 are presented. From the title of these publication it seems that the influence of high magnetic inductions above 1Tesla was applied.
Within the current work an induction of 350 micro-Tesla was applied – about a factor 10 higher than the static earth magnetic field, but several orders of magnitude smaller than in the cited publications.
I recommend adding a discussion on the influence of the field-strength.
Response 1(1): Thank you for your comments; it has now been elaborated in paper in accordance with your suggestion in the discussion session in resubmission.
Point 1(2): Time varying EMFs are not covered within the introduction. However, these effects are presented within the discussion chapter lines 227ff. To my point of view the lines 227ff should be shifted to the introduction part. The effect of heating by time varying magnetic fields depends on amplitude and frequency. Discussions of time varying EMFs on health features are going on.
Response 1(2): In the Introduction Section, the background of Time-varying EMF has now been further elaborated.
Point 2:Line 69, line 125ff: “Previous driving experience was not compulsory” – To my point of view driving skills depend on some experience. Was there an influence within the results of experienced drivers and less experienced ones?
According to the procedure (line 125ff) each subject performed the test under sham exposure first. Then it was repeated under field exposure. Assuming that the experience from the first run improves the subjects driving ability this could lead to a compensation of negative effects of the magnetic field.
Why is this assumption from me not correct?
According to the procedure (line 125ff) each subject performed the test under sham exposure first. Then it was repeated under field exposure. Assuming that the experience from the first run improves the subjects driving ability this could lead to a compensation of negative effects of the magnetic field.
Response 2: Thank you for your comments. . We have taken measures for ensuring the driving performance levels of subjects are consistent throughout the experiment. We would think different levels of driving experiences would not have a significant effort on the outcomes based in the following considerations:
The simulating driving system is of a simple composition, including a steering wheel and a pair of pedals for breaking and accelerating, with which is easy to familiar to operate.
Subjects were also provided with a driving practice on the simulating system prior the formal experiment, as mentioned to familiarize the operating system of the experiment in line 133 to 135.
To the experience in our observations during the preparation of the experiment, college students would have a quick-learning ability with their strong curiosity, and they would have a good driving performance and familiarization of the driving operating on the simulating system.
I hope the above would clarify your concern.
Point 3: Figure2: the green line in the legend cannot be seen very well. I recommend optimizing the figure.
Response 3: Thank you for your comments, the figure have now been improved accordingly.
Point 4: Lines 117 to 124 are repeated in lines 125 to 133.
Response 4: Thank you for spotting the repeated paragraph, the repeated paragraph has now been deleted.
Point 5: Line 157: “ … This awake stage of the driver in the experiment in corresponding to the inactive characteristics of the ...” maybe an “in” should be an “is”: “experiment is corresponding”
Response 5: The missing “is” has been added in the resubmission.
Point 6: Line 180 “…Figure 3, indicating that ...” – I guess it should be “table 3”
Response 6: Thank you; it has now been corrected.
Point 7: Line 222 ff: Here the authors present that in previous work inductions in the range of Tesla did not lead to “significant impact”.
I strongly recommend to shift the presentation of different field strength to the introduction and give a motivation why the authors investigated the small magnetic fields.
Response 7:
The presentation of different field strengths have been included in the introduction session. The motivation of 350 μT has been further presented in the Discussion Section.
Reviewer 3 Report
Dear authors,
Thank you for your courage to publish a negative result. More people should do that! You did a very thorough job in the statistical analysis. I do have some questions, and remarks, which you can find in the pdf. One thing that would certainly improve your paper is the conclusion. You say that the effect of the magnetic field is not significant. But you mean statistically significant, which is not the same as relevant. Based on your thorough statistical analysis, you would be able to say something like "The effect of the 0.35 mT static field on the reaction times of our 17 subjects is statistically not relevant. We must conclude that if there is any effect, it must be lower than x %". (If you would have measured millions of subject, you might have found a statistically significant effect of 0.01% for instance, which is not relevant I suppose).

Round 2
Reviewer 1 Report
I recommend the work for publication